# Hematological Events Potentially Associated with CDK4/6 Inhibitors: An Analysis from the European Spontaneous Adverse Event Reporting System

**DOI:** 10.3390/ph16101340

**Published:** 2023-09-22

**Authors:** Vera Martins, Mafalda Jesus, Luísa Pereira, Cristina Monteiro, Ana Paula Duarte, Manuel Morgado

**Affiliations:** 1Health Sciences Faculty, University of Beira Interior (FCS-UBI), 6200-506 Covilhã, Portugal; vera.isabel.martins@ubi.pt (V.M.); mafalda.jesus@ubi.pt (M.J.); csjmonte@ubi.pt (C.M.); apcd@ubi.pt (A.P.D.); 2Health Sciences Research Centre, University of Beira Interior (CICS-UBI), 6200-506 Covilhã, Portugal; 3CMA-UBI, Centre of Mathematics and Applications, University of Beira Interior, Rua Marquês d’Ávila e Bolama, 6201-001 Covilhã, Portugal; lpereira@ubi.pt; 4UFBI—Pharmacovigilance Unit of Beira Interior, University of Beira Interior, 6200-506 Covilhã, Portugal; 5Pharmaceutical Services, University Hospital Center of Cova da Beira, 6200-251 Covilhã, Portugal

**Keywords:** CDK4/6 inhibitors, abemaciclib, palbociclib, ribociclib, hematological adverse drug reactions, leukopenia, thrombocytopenia, EudraVigilance, breast cancer

## Abstract

Cyclin-dependent kinases 4 and 6 (CDK4/6) inhibitors are a recent targeted therapy approved for patients with hormone receptor-positive (HR+), human epidermal growth factor receptor 2 negative (HER2−) advanced breast cancer. Abemaciclib, palbociclib and ribociclib demonstrated great efficacy and safety during clinical studies. However, differences in their adverse-event profiles have been observed. This work aims to describe the suspected adverse drug reactions (ADRs), such as leukopenia and thrombocytopenia, reported for each CDK4/6 inhibitor in the EudraVigilance (EV) database. Data on individual case safety reports (ICSRs) were obtained by accessing the European spontaneous reporting system via the EV website. Information on concomitant drug therapy, including fulvestrant, letrozole, anastrozole and exemestane, was also analyzed. A total of 1611 ICSRs were collected from the EV database. Most reports of palbociclib and ribociclib were classified as serious cases for both suspected leukopenia and thrombocytopenia ADRs. However, most patients had their leukopenia and thrombocytopenia recovered/resolved. On the contrary, reports of abemaciclib were mostly characterized as non-serious cases. Abemaciclib and palbociclib were often combined with fulvestrant, while ribociclib was generally associated with letrozole. Pharmacovigilance studies are crucial for the early identification of potential ADRs and to better differentiate the toxicity profile of the different CDK4/6 inhibitors, particularly in a real-world setting.

## 1. Introduction

Hormone receptor-positive (HR+)/human epidermal growth factor receptor 2 negative (HER2−) tumors represent the most common breast cancer subtype [1,2]. Patients with de novo or relapsed HR+ advanced breast cancer require effective treatment to delay disease progression. In this context, the traditional approach considered has been the use of endocrine therapy (ET) as a single agent [3,4]. ET can decrease the rate of recurrence at an early-stage disease; however, ET resistance mechanisms turn out to be a major oncology concern [3,5,6]. This dilemma highlighted the importance of new treatment options, especially based on targeted antineoplastic therapies [3,7].

On 15 September 2016, the European Medicines Agency’s (EMA) Committee for Medicinal Products for Human Use (CHMP) recommended a marketing authorization for the first inhibitor of cyclin-dependent kinases 4 and 6 (CDK4/6) named palbociclib [8]. Palbociclib was approved for women with locally advanced or metastatic breast cancer with estrogen receptor-positive (ER+)/HER2− subtype in combination as first-line therapy with aromatase inhibitors and as second-line therapy in combination with fulvestrant, particularly in women who have received prior ET [9,10]. Later, two other CDK4/6 inhibitors were approved for the same clinical indication in the European Union market, namely ribociclib and abemaciclib [10,11,12]. CDK4/6 inhibitors induce their anti-cancer activity through the CD4/6-Rb axis, which is often disrupted in the majority of cancers and is at the basis of abnormal cell proliferation. When DNA synthesis occurs, CDK4/6 kinases, in the presence of mitogenic signals, bind to the D-type cyclins (cyclin D1, cyclin D2 and cyclin D3) and form catalytically active complexes. One key function of these complexes is associated with the phosphorylation of the retinoblastoma tumor suppressor protein (RB). This protein is responsible for the cell cycle transition from G1 to S phase. By preventing RB phosphorylation, CDK4/6 inhibitors block cell cycle progression [13,14]. Structurally, ribociclib is very similar to palbociclib. Nevertheless, abemaciclib is the least similar in terms of structure [15], presenting several differences from the other inhibitors. In fact, abemaciclib is the only inhibitor granted by the U.S. Food and Drug Administration (FDA) to be used as a monotherapy agent [16,17].

Double-blind randomized clinical trials PALOMA-2 (NCT01740427) and PALOMA-3 (NCT01942135) were crucial to collecting the efficacy and safety data of palbociclib. Both trials demonstrated that when palbociclib is used in combination with an aromatase inhibitor (letrozole) or fulvestrant, it significantly enhanced progression-free survival (PFS) value versus those women treated with letrozole or fulvestrant alone [18,19]. Other clinical trials, such as MONALEESA and MONARCH also contributed significant data on the efficacy of ribociclib and abemaciclib, respectively [20,21,22,23]. Although the drug combinations tested seem to be safe and effective, palbociclib is associated with higher rates of hematologic adverse events (AEs), especially neutropenia, when compared to ET alone [24]. Costa et al. performed a meta-analysis study of patients taking palbociclib or ribociclib [25]. The results reported that the patients involved had an absolute risk of grade 3/4 neutropenia of 61% and a risk of grade 3/4 leukopenia of 25% [25]. In addition, Kassem et al. reported a study that in the CDK4/6 inhibitors arm, the incidence of all-grade leukopenia ranged from 20.8 to 45.5% and all-grade thrombocytopenia from 9 to 36.2% [26]. Later, a systematic review and meta-analysis study suggested several differences in the toxicity profile between CDK4/6 inhibitors. In this context, palbociclib and ribociclib showed high rates of hematological toxicity. However, abemaciclib was more associated with a high rate of gastrointestinal toxicities [27].

Considering that this class of drugs is relatively recent in the clinical oncology practice and that there is still very limited information in the real-world setting about the toxicity profile of these agents, the present pharmacovigilance study aims to analyze two of the most suspected hematological adverse drug reactions (ADRs)—leukopenia and thrombocytopenia—reported in the European EudraVigilance (EV) database associated to CDK4/6 inhibitors—palbociclib, ribociclib and abemaciclib. In addition, we also intended to identify the most frequent concomitant drugs mentioned in the individual case safety reports (ICSRs) of CDK4/6 inhibitors.

## 2. Results

### 2.1. Demographic Characteristics of ICSRs Regarding Leukopenia Reports

A total of 822 ICSRs have been reported for leukopenia as suspected ADR and mentioning one CDK4/6 inhibitor as a suspected drug. Most of these reports were associated with palbociclib with a total of 586 ICSRs; 85 ICSRs were related to abemaciclib and 151 ICSRs pertained to ribociclib. Transversally to all CDK4/6 inhibitors and as expected, considering the therapeutic indications of these drugs, females were the most frequent patients with a total of 792 ICSRs (96.35%). Only 10 ICSRs (1.22%) were associated with male patients. Adults (18–64 years) and elderly patients (65–85 years) were the two most referred age groups, with 357 (43.43%) and 278 (33.82%) ICSRs, respectively. In addition, a significant number of ICSRs were considered “Not specified” in terms of age group (N = 168, 20.44%). Most ICSRs were reported by a healthcare professional (N = 712, 86.62%) and came from the Non-European Economic Area (N = 512, 62.29%). Concerning the outcome, more than half of ICSRs were classified as “unknown” outcomes (N = 485, 59.0%) and 158 ICSRs (19.22%) reported that leukopenia was recovered or resolved. Regarding the seriousness of the reported cases, a high percentage was classified as a serious case (N = 760, 92.46%) with the criterion of “Other medically important information” (N = 740, 90.02%). Abemaciclib was the only inhibitor that presented a greater number of non-serious leukopenia reports (N = 51, 60.0%), while the majority of palbociclib (N = 579, 98.81%) and ribociclib (N = 147, 97.35%) reports were considered as serious. Also, it is important to highlight that a total of 7 fatal cases were associated with palbociclib and 21 cases required/prolonged hospitalization. Concerning the concomitant therapy, the majority of ICSRs did not include fulvestrant, letrozole, anastrozole or exemestane as concomitant therapy (N = 455, 55.35%). In fact, most of the reports did not include any concomitant therapy. A total of 367 (44.65%) leukopenia reports mentioned, at least, one of the following concomitant medicines: fulvestrant, letrozole, anastrozole and exemestane. The most frequent concomitant drug found in ICSRs reports was letrozole (N = 179, 21.78%), followed by fulvestrant (N = 162, 19.71%). Both palbociclib (N = 139, 23.72%) and ribociclib (N = 22, 14.57%) frequently appeared associated with letrozole. Exemestane was the least reported concomitant agent, appearing in only nine reported cases of leukopenia (1.09%). There were no reports for ribociclib plus exemestane. These results are presented in Table 1 and in Figure 1.

### 2.2. Demographic Characteristics of ICSRs Regarding Thrombocytopenia Reports

Concerning suspected thrombocytopenia reports, a total of 789 ICSRs were retrieved from the EV platform, with 121 associated with abemaciclib, 591 with palbociclib and 77 with ribociclib. Similar to leukopenia reports, female was the most common gender reported, counting with 740 (93.79%) cases in total. In this context, in males, all of the reports were associated with palbociclib (2.54%) as the suspected drug. The elderly population was the most referred age group among CDK4/6 inhibitors reports (N = 351, 44.49%). Most thrombocytopenia ICSRs were reported by a healthcare professional (N = 735, 93.16%) and based in the European Economic Area (N = 455, 57.67%). Regarding the outcome section, those defined as “unknown” were the most frequent (N = 417, 52.85%), followed by “recovered/resolved” reports (N = 189, 23.95%). Many thrombocytopenia reports were considered serious (N = 678, 85.93%), and just 111 ICSRs (14.07%) were classified as non-serious. Abemaciclib presented a slightly higher number of non-serious cases (N = 65, 53.72%) compared to the serious ones (N = 56, 46.28%). A total of 6 (0.76%) fatal events were notified in the outcome section. However, 15 (1.90%) deaths appeared in the seriousness criteria. Palbociclib was related to 12 (2.03%) deaths, abemaciclib to 1 (0.83%) and ribociclib to 2 (2.60%). In addition, a total of 13 reports led to life-threatening conditions and 74 cases of suspected thrombocytopenia required/prolonged hospitalization. Regarding concomitant therapy, fulvestrant was still the most commonly reported drug for abemaciclib (N = 29, 23.97%) and letrozole for ribociclib (N = 15, 19.48%). Palbociclib showed 148 (25.04%) reports, including fulvestrant, and a fewer number of cases, including letrozole (N = 108, 18.27%). Anastrozole and exemestane were reported less often as concomitant therapeutic agents (N = 17, 2.15% versus N = 12, 1.52%, respectively). However, the number of reports with no concomitant medicines (57.03%) was slightly superior to ICSRs that reported concomitant medicines (42.97). These results are presented in Table 2 and in Figure 2.

The seriousness of adverse drug reactions (leucopenia and thrombocytopenia) associated with CDK4/6 inhibitors (whether as monotherapy or in combination with concomitant drug therapy) was explored through a multivariate logistic regression (Table 3). Serious cases of leukopenia and thrombocytopenia were significantly more likely to occur for palbociclib and ribociclib when compared to abemaciclib (leukopenia–palbociclib: OR = 146.220, 95% CI: 58.38–360.08; ribociclib: OR = 50.568, 95% CI: 16.85–151.81; thrombocytopenia—palbociclib: OR = 14.690, 95% CI: 9.13–23.63; ribociclib: OR = 30.220, 95% CI: 8.99–101.57). When a CDK4/6 inhibitor was used in combination with another drug therapy (fulvestrant, letrozole, anastrozole, or exemestane) the odds of serious cases of leukopenia decreased (OR = 0.294, 95% CI: 0.14–0.62).

## 3. Discussion

CDK4/6 inhibitors are a recent pharmacological class of drugs in breast cancer treatment, showing promising efficacy and relevant safety results [27,28]. Clinical trials such as PALOMA, MONARCH and MONALEESA were crucial to collect initial clinical data concerning CDK4/6 inhibitors, especially in relation to their association with an aromatase inhibitor (letrozole, anastrozole or exemestane) or fulvestrant [29]. In this context, the group treated with a CDK4/6 inhibitor presented longer PFS than the placebo group [1,19,20,21,22,23,30,31], highlighting the promising results of this class of drugs. This study intended to investigate leukopenia and thrombocytopenia spontaneous reports related to the approved CDK4/6 inhibitors, palbociclib, ribociclib, and abemaciclib, through the analysis of data provided by EV.

A total of 1611 ICSRs were extracted, taking into consideration the date of marketing authorization granted by EMA for the last inhibitor to be approved, abemaciclib. All ICSRs involving suspected leukopenia and thrombocytopenia were considered from 1 January 2018 to 31 December 2022. In this context, 822 ICSRs reported leukopenia as a suspected ADR and 789 ICSRs of suspected thrombocytopenia were analyzed. As expected, female patients were the main population reported by healthcare professionals, especially in the age groups of 18–64 years and 65–85 years old. Only 25 male reports (10 ICSRs for leukopenia and 15 ICSRs for thrombocytopenia) were reported. This fact can be explained considering the approved therapeutic indications of CDK4/6 inhibitors and considering that breast cancer is much more common in women than in men. All CDK4/6 inhibitors were approved for metastatic HR+/HER2−negative breast cancer [9,11,12]. Also, according to Allen et al., male breast cancer is rare, representing less than 1% of all breast cancer cases. This evidence can be correlated with the lower incidence of predisposing risk factors men face when compared to women population [32]. Additionally, women are generally more prone to drug-induced adverse events based on lower lean body mass, reduced hepatic clearance, and differences in enzyme activities (e.g., cytochrome P450) [33,34].

In our study, palbociclib presented the highest number of suspected ADRs of leukopenia and thrombocytopenia (586 ICSRs versus 591 ICSRs, respectively). In addition, ribociclib presented a higher number of suspected leukopenia cases than abemaciclib (151 ICSRs versus 85 ICSRs, respectively). On the other hand, suspected thrombocytopenia was slightly more reported in patients taking abemaciclib than ribociclib (121 ICSRs versus 77 ICSRs). A multivariate logistic regression analysis of these suspected hematologic ADRs revealed that serious cases of leukopenia and thrombocytopenia were significantly more likely to occur for palbociclib (leukopenia—OR = 146.220, 95% CI: 58.38–360.08; thrombocytopenia—OR = 14.690, 95% CI: 9.13–23.63) and ribociclib (leukopenia—OR = 50.568, 95% CI%: 16.85–151.81; thrombocytopenia—OR = 30.220, 95% CI: 8.99–101.57) when compared to abemaciclib. When a CDK4/6 inhibitor was used in combination with another drug therapy (fulvestrant, letrozole, anastrozole, exemestane) the odds of serious cases of leukopenia decreased (OR = 0.294, 95% CI: 0.14–0.62) when compared to monotherapy. However, we consider that additional studies are necessary to confirm and/or clarify these findings.

Hematologic disturbs are the most common toxicities reported among CDK4/6 inhibitors, namely in palbociclib and ribociclib [27,35]. This fact is related to their action on CDK6, which is a key regulator of hematopoietic precursor proliferation [36]. According to the Summary of Product Characteristics (SmPC) of CDK4/6 inhibitors, leukopenia is considered a very common reaction in patients taking palbociclib, ribociclib and abemaciclib [9,11,12]. Thrombocytopenia is also classified as a very common reaction (≥1/10) for palbociclib and abemaciclib. However, ribociclib is only classified as a common reaction (≥1/100 to <1/10) [9,11,12]. Onesti et al. explored the safety profiles of CDK4/6 inhibitors through a systematic review and meta-analysis. In this study, palbociclib and ribociclib showed a high rate of neutropenia, leukopenia, anemia, and thrombocytopenia; however, for abemaciclib, gastrointestinal toxicities were the most reported ADRs, including diarrhea, nausea, decreased appetite, and abdominal pain [27]. In general, gastrointestinal disorders are the most frequent events for abemaciclib, and for this reason, this CDK4/6 inhibitor should not be recommended in patients with gastrointestinal comorbidities [37]. Considering the information described above, it is possible to verify that although their mechanisms of action and efficacy are similar, some differences can be found in their toxicity profiles.

Concerning the outcome of leukopenia and thrombocytopenia, most cases were classified as recovered/resolved. In fact, these results may be verified considering the therapeutic regimen of palbociclib and ribociclib. Palbociclib and ribociclib are administered for three consecutive weeks, followed by a week’s break. This break allows the recovery of hematopoietic progenitors [38]. Abemaciclib has a higher affinity for CDK4 with a lower IC50 and can therefore be administered continuously, presenting less hematopoietic toxicity than palbociclib and ribociclib [27]. In fact, according to our results, abemaciclib showed a higher percentage of non-serious cases of suspected leukopenia and thrombocytopenia. However, it should be noted that more than half of the cases were reported with an “Unknown” outcome.

Regarding the seriousness of collected ICSRs for palbociclib and ribociclib, most reports were considered as serious and classified with the seriousness criteria “Other medically important information”. In addition, 7 deaths were associated with the suspected ADR leukopenia, particularly with the inhibitor palbociclib, and 15 deaths were associated with the suspected ADR thrombocytopenia, of which 12 are again associated with palbociclib. Diéras et al. evaluated the safety of palbociclib based on PALOMA trials. In the three pooled PALOMA studies, no deaths occurred during the trials or the 28 days after the last dose administered, among patients receiving palbociclib [39]. In this case, we believe the casual relationship between ADRs and the suspected drugs, particularly palbociclib, should be evaluated and established. It is also important to highlight that the number of deaths reported in the “seriousness criteria” for both leukopenia and thrombocytopenia reports is not the same as the number of deaths reported in the “outcome”. The discrepancies may be related to the existence of some errors in completing or updating ADR reports. These results reinforce the need for reports of suspected ADRs to be as complete as possible, as this is essential for a proper assessment of causality [40].

Letrozole and fulvestrant were considered the main concomitant therapy associated with CDK4/6 inhibitors both in suspected leukopenia and thrombocytopenia reports. The combination of endocrine therapy with CDK4/6 inhibitors has been approved and extensively described in the literature [41]. Other combinations, such as with anastrozole and exemestane, may also occur, revealing better results than ET alone [42].

Although CDK4/6 inhibitors are generally safe and manageable drugs, differences in their toxicity profile may lead to different clinical choices. Pharmacovigilance is an extremely important tool for the early detection of potentially relevant ADRs, contributing to a more responsible use of medication and reducing the burden on healthcare systems [43].

For this study, some strengths and limitations should be considered. The access to EV, a large and comprehensive spontaneous ADR reports database, is considered the major strength of this article. CDK4/6 inhibitors are a recent class of antineoplastic drugs, and, in this context, most safety data currently available were obtained from clinical trials. Through the EV database, we were able to analyze ICSRs from a heterogeneous population in a real-world setting and better characterize two of the main hematologic ADRs of CDK4/6 inhibitors. Despite our efforts to minimize bias, there are important limitations that should be highlighted. Crucial information on ICSRs was missing, such as ADR outcomes and concomitant medications (the fact that no concomitant medication was indicated in the report does not mean that the patient did not take it; it just may not have been reported). In fact, a significant percentage of suspected leukopenia and thrombocytopenia reports was reported with an “Unknown” outcome. In addition, some ICSRs reported death as a seriousness criterion. However, in the outcome section, the same suspected ADRs (in the same ICSRs) were not reported as fatal in the outcome section. The phenomenon of underreporting and underestimation of the frequency of ADRs in oncology should be considered [43]. Finally, it is important to mention that this data does not provide evidence of the causality relationship between the analyzed ADRs and the suspected drugs.

## 4. Materials and Methods

### 4.1. Data Source

EV is the system responsible for collecting, managing, and analyzing information on suspected adverse reactions to medicines authorized in the European Economic Area (EEA) [44]. This platform is operated by EMA and is considered one of the heartwoods of pharmacovigilance department, having an important role in the early detection and management of risks related to the use of medicines and vaccines [45].

Data on ICSRs were extracted from the website of suspected ADR of the EV database by accessing www.adrreports.eu (accessed on 1 June 2023). The ADRs included in each ICSR are coded according to the Medical Dictionary for Regulatory Activities (MedDRA) terminology (https://www.meddra.org, accessed on 1 June 2023). MedDRA is a rich and highly specific standardized medical terminology to facilitate the international sharing of regulatory information for medical products used by humans.

### 4.2. Data Selection

A retrospective and descriptive study was performed using information on spontaneous reports submitted to the EV database. All ICSRs with a CDK4/6 inhibitor—palbociclib, ribociclib or abemaciclib—as a suspected drug were considered in the period from 1 January 2018 to 31 December 2022. This period was selected based on the most recent authorization year for a CDK4/6 inhibitor granted by EMA. The date of issue of marketing authorization valid throughout the European Union for abemaciclib was 26 September 2018.

Open access data were used. Thus, no access authorization was needed.

#### Individual Cases Safety Reports Extraction and Descriptive Analysis

In the EV database, by using the line listing function, we selected all ICSRs with leukopenia and thrombocytopenia as reported suspected reactions from 1 January 2018 to 31 December 2022. Note that, on the tab “Reaction groups”, System Organ Class (SOC) “Blood and lymphatic system disorders” was selected. The CDK4/6 inhibitors considered were abemaciclib, palbociclib and ribociclib. Information was collected on sex, age group, reporter group, geographic origin, ADR outcome, seriousness, and seriousness criteria. According to the International Council on Harmonization E2D (Post-Approval Safety Data Management: Definition and Standards for Expedited Reporting E2D) guideline, a case is classified as “serious” if a congenital anomaly/birth defect is detected, results in disability or incapacity, becomes life-threatening, results in death, requires or prolongs hospitalization, or results in another significant clinical condition [46]. In addition, for the ICSRs that reported two or more outcomes for the same hematological event, we considered the outcome with the highest level of resolution. Also, information on concomitant therapy was retrieved from each ICSR. All suspected ADR reports in which CDK4/6 inhibitors were not described as the only suspected drug were excluded.

Using Office^®^ Excel^®^ 365 software, Version 2211 (Microsoft Corporation, Redmond, WA, USA), categorical variables were described and analyzed through their absolute and relative frequency.

### 4.3. Statistical Analysis

Categorical variables were presented with their relative and absolute values. For statistical analysis, the Qui-square test was applied using the statistical software package SPSS 28.0 for Windows (SPSS. Chicago, IL, USA). A multivariate logistic regression analysis was performed to determine independent risk factors for the seriousness of leukopenia and thrombocytopenia. *p* < 0.05 indicates a statistically significant test result.

## 5. Conclusions

Spontaneous reports of CDK4/6 inhibitors related to hematological events—leukopenia and thrombocytopenia—were analyzed through a retrospective study conducted on data retrieved from the EV database. CDK4/6 inhibitors, including palbociclib, rubociclib and abemaciclib, are a recently targeted class of antineoplastic drugs, and, although their mechanism of action and efficacy are similar, some differences can be found in their toxicity profiles. It is important for health professionals to be aware of these different toxicity profiles associated with CDK4/6 inhibitors in a real-world setting, with a view to developing strategies for safer and more effective prescription and use of these drugs. More high-quality studies should be conducted on this class of drugs in order to better establish their toxicity profiles in a real-world setting.

## Figures and Tables

**Figure 1 pharmaceuticals-16-01340-f001:**
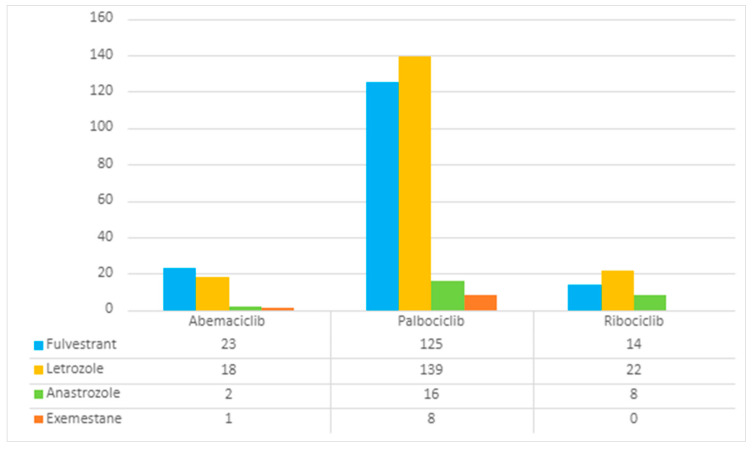
Comparative analysis of concomitant therapy appearing in reports (ICSRs) of suspected leukopenia for CDK4/6 inhibitors.

**Figure 2 pharmaceuticals-16-01340-f002:**
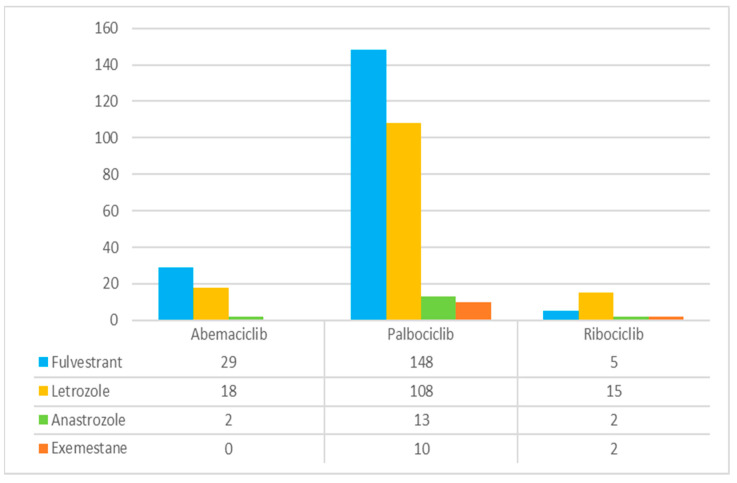
Comparative analysis of concomitant therapy appearing in reports of suspected thrombocytopenia for CDK4/6 inhibitors.

**Table 1 pharmaceuticals-16-01340-t001:** Demographic characteristics of ICSRs of suspected leukopenia reports involving abemaciclib, palbociclib or ribociclib reported in the EV database from 1 January 2018 to 31 December 2022.

	Individual Case Safety Reports (%)				
	Abemaciclib N = 85	Palbociclib N = 586	Ribociclib N = 151	Total N = 822	*p*
Sex					0.78
Male	1 (1.18)	8 (1.37)	1 (0.66)	10 (1.22)	
Female	81 (95.29)	566 (96.59)	145 (96.03)	792 (96.35)	
Not Specified	3 (3.53)	12 (2.05)	5 (3.31)	20 (2.43)	
Age Group					<0.001
Pediatrics (<18 years)	1 (1.18) ^a^	1 (0.17) ^a^	1 (0.66) ^a^	3 (0.36)	
Adult (18–64 years)	40 (47.06) ^a^	252 (43.0) ^a^	65 (43.05) ^a^	357 (43.43)	
Eldery (65–85 years)	25 (29.41) ^a,b^	228 (38.91) ^b^	25 (16.56) ^a^	278 (33.82)	
Very Eldery (>85 years)	1 (1.18) ^a^	14 (2.39) ^a^	1 (0.66) ^a^	16 (1.95)	
Not Specified	18 (21.18) ^a^	91 (15.53) ^a^	59 (39.07) ^b^	168 (20.44)	
Type of Reporter					<0.001
Health Care Professional	83 (97.65) ^a^	481 (82.08) ^b^	148 (98.01) ^a^	712 (86.62)	
Non-Health Care Professional	2 (2.35) ^a^	105 (17.92) ^b^	3 (1.99) ^a^	110 (13.38)	
Region					<0.001
European Economic Area	60 (70.59) ^a^	154 (26.28) ^b^	95 (62.91) ^a^	309 (37.59)	
Non-European Economic Area	25 (29.41) ^a^	432 (73.72) ^b^	55 (36.42) ^a^	512 (62.29)	
Not Specified	0 ^a^	0 ^a^	1 (0.66) ^a^	1 (0.12)	
Outcome					<0.001
Recovered/Resolved	11 (12.94) ^a^	86 (14.68) ^a^	61 (40.40) ^b^	158 (19.22)	
Recovered/Resolved with Sequelae	0 ^a^	0 ^a^	2 (1.32) ^b^	2 (0.24)	
Recovering/Resolving	19 (22.35) ^a^	57 (9.73) ^b^	18 (11.92) ^b^	94 (11.44)	
Not Recovered/Not Resolved	2 (2.35) ^a^	70 (11.95) ^b^	11 (7.28) ^a,b^	83 (10.10)	
Fatal	0	0	0	0	
Unknown	53 (62.35) ^a^	373 (63.65) ^a^	59 (39.07) ^b^	485 (59.0)	
Seriousness					<0.001
Non-Serious	51 (60.0) ^a^	7 (1.19) ^b^	4 (2.65) ^b^	62 (7.54)	
Serious	34 (40.0) ^a^	579 (98.81) ^b^	147 (97.35) ^b^	760 (92.46)	
Seriousness Criteria					
Other (other medically important information)	25 (29.41) ^a^	573 (97.78) ^b^	142 (94.04) ^b^	740 (90.02)	<0.001
Congenital	0	0	0	0	
Disability	0	1 (0.17)	0	1 (0.12)	0.817
Hospitalization	12 (0.14) ^a^	21 (3.58) ^b^	5 (3.31) ^b^	38 (4.62)	<0.001
Life-threatening	1 (1.18)	4 (0.68)	2 (1.32)	7 (0.85)	0.703
Death	0	7 (1.19)	0	7 (0.85)	0.241
Concomitant Therapy					<0.001
Yes	44 (51.76) ^a^	280 (47.78) ^a^	43 (28.48) ^b^	367 (44.65)	
No	41 (48.24) ^a^	306 (52.22) ^a^	108 (71.52) ^b^	455 (55.35)	
Concomitant Therapy					
Fulvestrant	23 (27.06) ^a^	125 (21.33) ^a^	14 (9.27) ^b^	162 (19.71)	<0.001
Letrozole	18 (21.18)	139 (23.72)	22 (14.57)	179 (21.78)	0.052
Anastrozole	2 (2.35)	16 (2.73)	8 (5.30)	26 (3.16)	0.268
Exemestane	1 (1.18)	8 (1.37)	0	9 (1.09)	0.355

Percentages in columns with different letters are significantly different (*p* < 0.05).

**Table 2 pharmaceuticals-16-01340-t002:** Demographic characteristics of ICSRs of suspected thrombocytopenia reports involving abemaciclib, palbociclib and ribociclib reported in the EV database from 1 January 2018 to 31 December 2022.

	Individual Case Safety Reports (%)				
	Abemaciclib N = 121	Palbociclib N = 591	Ribociclib N = 77	Total N = 789	*p*
Sex					0.103
Male	0	15 (2.54)	0	15 (1.90)	
Female	117 (96.69)	552 (93.40)	71 (92.21)	740 (93.79)	
Not Specified	4 (3.31)	24 (4.06)	6 (7.79)	34 (4.31)	
Age Group					0.032
Pediatrics (<18 years)	0	1 (0.17)	0	1 (0.13)	
Adult (18–64 years)	43 (35.54)	203 (34.35)	23 (29.87)	269 (34.09)	
Elderly (65–85 years)	47 (38.84)	272 (46.02)	32 (41.56)	351 (44.49)	
Very Elderly (>85 years)	1 (0.83)	22 (3.72)	0	23 (2.92)	
Not Specified	30 (24.79)	93 (15.74)	22 (28.57)	145 (18.38)	
Type of Reporter					0.021
Health Care Professional	118 (97.52) ^a^	542 (91.71) ^b^	75 (97.40) ^a^	735 (93.16)	
Non-Health Care Professional	3 (2.48) ^a^	49 (8.29) ^b^	2 (2.60) ^a^	54 (6.84)	
Region					0.014
European Economic Area	87 (71.90) ^a^	323 (54.65) ^b^	45 (58.44) ^b^	455 (57.67)	
Non-European Economic Area	34 (28.10) ^a^	267 (45.18) ^b^	32 (41.56) ^b^	333 (42.21)	
Not Specified	0 ^a^	1 (0.17) ^a^	0 ^a^	1 (0.13)	
Outcome					0.082
Recovered/Resolved	26 (21.49) ^a^	146 (24.70) ^a^	17 (22.08) ^a^	189 (23.95)	
Recovered/Resolved with Sequelae	1 (0.83) ^a^	3 (0.51) ^a^	0 ^a^	4 (0.51)	
Recovering/Resolving	20 (16.53) ^a^	52 (8.80) ^b^	4 (5.19) ^b^	76 (9.63)	
Not Recovered/Not Resolved	19 (15.70) ^a^	68 (11.51) ^a^	10 (12.99) ^a^	97 (12.29)	
Fatal	0 ^a^	4 (0.68) ^a^	2 (2.60) ^a^	6 (0.76)	
Unknown	55 (45.45)	318 (53.81)	44 (57.14)	417 (52.85)	
Seriousness					<0.001
Non-serious	65 (53.72) ^a^	43 (7.28) ^b^	3 (3.90) ^b^	111 (14.07)	
Serious	56 (46.28) ^a^	548 (92.72) ^b^	74 (96.10) ^b^	678 (85.93)	
Seriousness Criteria					
Other (other medically important information)	36 (29.75) ^a^	522 (88.32) ^b^	68 (88.31) ^b^	626 (79.34)	<0.001
Congenital	0	0	0	0	
Disability	0	1 (0.17)	0	1 (0.13)	0.846
Hospitalization	21 (17.36) ^a^	48 (8.12) ^b^	5 (6.49) ^b^	74 (9.38)	0.004
Life-threatening	4 (3.31)	7 (1.18)	2 (2.60)	13 (1.65)	0.196
Death	1 (0.83)	12 (2.03)	2 (2.60)	15 (1.90)	0.606
Concomitant Therapy					0.037
Yes	47 (38.84) ^a,b^	268 (45.35) ^a^	24 (31.17) ^b^	339 (42.97)	
No	74 (61.16) ^a,b^	323 (54.65) ^a^	53 (68.83) ^b^	450 (57.03)	
Concomitant Therapy					
Fulvestrant	29 (23.97) ^a^	148 (25.04) ^a^	5 (6.49) ^b^	182 (23.07)	0.001
Letrozole	18 (14.88)	108 (18.27)	15 (19.48)	141 (17.87)	0.625
Anastrozole	2 (1.65)	13 (2.20)	2 (2.60)	17 (2.15)	0.895
Exemestane	0	10 (1.69)	2 (2.60)	12 (1.52)	0.275

Percentages in columns with different letters are significantly different (*p* < 0.05).

**Table 3 pharmaceuticals-16-01340-t003:** Multivariate logistic regression analysis of the relationship between the seriousness of reported cases of leucopenia and thrombocytopenia and CDK4/6 inhibitors as well as concomitant drug therapy.

	Adverse Drug Reactions—Individual Case Safety Reports					
	Leukopenia			Thrombocytopenia		
	Serious	Non Serious	*p* Value OR (95% CI)	Serious	Non Serious	*p* Value OR (95% CI)
Drug			<0.001			<0.001
*Abemaciclib*	34	51	Ref.	56	65	Ref.
*Palbociclib*	579	7	146.220	548	43	14.690
			(58.38–360.08)			(9.13–23.63)
*Ribociclib*	147	4	50.568	74	3	30.220
			(16.85–151.81)			(8.99–101.57)
Concomitant therapy			0.001			0.069
No	433	22	Ref.	377	73	Ref.
Yes	327	40	0.294	301	38	1.559
			(0.14–0.62)			(0.967–2.516)

CI—confidence interval, OR—odds ratio, Ref.—reference drug for statistical analysis.

## Data Availability

All data the authors extracted from the European Medicines Agency (EMA) pharmacovigilance database called EudraVigilance are publicly available. Pharmacovigilance data from EudraVigilance are publicly available at www.adrreports.eu (accessed on 5 June 2023).

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
