# Peer review of "Hematological Events Potentially Associated with CDK4/6 Inhibitors: An Analysis from the European Spontaneous Adverse Event Reporting System"

_pharmaceuticals, 2023, doi:10.3390/ph16101340_

Round 1

Reviewer 1 Report

It is a quite well written study regarding CDK4/6 inhibitors. It suits to the profil of this journal. It contains already known knowledge but given with clarity.

Introduction mention all the studies that have already taken place for the effectiveness bur for the safety also of CDK4/6 inhibitors.

In methods clearly emphasize the collection of the data from the European system.

In results in the tables and the figures all the data are analyzed when CDK4/6 inhibitors are given as minotherapy or in combination with other agents.

In discussion the review of the literature is extensive but also the advantages and the disadvantages of the present study through the European database.

The authors by themselves highlight the inadequate register of the information of the cases. The deaths regarding neutropenia and especially thrombocytopenia are not explained sufficiently. The fatal cases and the deaths are not in harmony.

In my opinion everything in this study has already known by previous studies and redarding each CDK4/6 inhibitor and their combination with other agents.

In present but also in futures studies CDK4/6 inhibitors are approved for other types of cancer than breast cancer and with other combination agents. Haematological events in these cases would have major interest in the future.

Minor editing in English language

Author Response

Responses to Reviewers

The authors are grateful for comments and suggestions from all reviewers. All comments and suggestions have been taken into account, which has helped to improve the quality of the article. All changes to the article are marked in yellow.

The English has also been revised and improved. All these English corrections have not been reported so as not to unnecessarily burden the revised article with corrections.

Point-to-point responses to Reviewers

Reviewer 1

It is a quite well written study regarding CDK4/6 inhibitors. It suits to the profile of this journal. It contains already known knowledge but given with clarity.

Ans: The authors are pleased with the interest the study has generated and are grateful for the summary of the main aspects of the article and the comments provided.

In my opinion everything in this study has already known by previous studies and redarding each CDK4/6 inhibitor and their combination with other agents.

Ans: Considering that this class of drugs (CDK4/6 inhibitors) is relatively recent in the clinical oncology practice and that there is still very limited information in the real-world setting about the toxicity profile of these agents, the present pharmacovigilance study constitutes a contribution to better understand the hematological toxicity (namely leukopenia and thrombocytopenia) of these drugs in the real-world setting (as mentioned in Introduction, first sentence of the last paragraph).

It is important for health professionals to be aware of these different toxicity profiles associated with CDK4/6 inhibitors in a real-world setting, with a view to developing strategies for safer and more effective prescription and use of these drugs ((as mentioned in Conclusions, penultimate sentence).

It should also be noted that the Methods, Results and Discussion were improved by carrying out a statistical analysis, using logistic regression, to determine independent risk factors for seriousness to leukopenia and thrombocytopenia.

Reviewer 2 Report

In this article, Martins et al. have investigated characteristics of patients with solid tumors treated with CDK4/6 inhibitors who have reported hematological toxicities, especially leukopenia and thrombocytopenia.

Few points should be addressed.

1. The Authors should discuss the probability that gender differences in reported AEs is due to the type of diagnosis and drug indication. Therefore, this finding should not be so emphasized, except if compared to a case control study.

2. Tables 1 and 2 and Figures 1 and 2 could benefit of p values.

3. A multivariate analysis should be performed.

Minor spelling

Author Response

Responses to Reviewers

The authors are grateful for comments and suggestions from all reviewers. All comments and suggestions have been taken into account, which has helped to improve the quality of the article. All changes to the article are marked in yellow.

The English has also been revised and improved. All these English corrections have not been reported so as not to unnecessarily burden the revised article with corrections.

Point-to-point responses to Reviewers

Reviewer 2

  1. The Authors should discuss the probability that gender differences in reported AEs is due to the type of diagnosis and drug indication. Therefore, this finding should not be so emphasized, except if compared to a case control study.

Ans: The authors agree with this comment and suggestion which contributed to improve the quality of the article.

The authors added the following in “Results”: (of the revised manuscript)

Transversally to all CDK4/6 inhibitors and as expected, considering the therapeutic indications of these drugs, female were the most frequent patients with a total of 792 ICSRs (96.35%).

The authors also added the following sentences in “Discussion”:

As expected, female patients were the main population reported by healthcare professionals, especially in the age groups of 18-64 years and 65-85 years old. Only 25 male reports (10 ICSRs for leukopenia and 15 ICSRs for thrombocytopenia) were reported. This fact can be explained considering the approved therapeutic indications of CDK4/6 inhibitors and considering that breast cancer is much more common in women than in men.

  1. Tables 1 and 2 and Figures 1 and 2 could benefit of p values.

Ans: The authors are grateful for this very pertinent suggestion that contributed to improving the quality of the article. In this way, the authors applied the Qui-square test to statistically analyse the categorical variables. The authors added the p-values were to Tables 1 and 2, having also discriminated which results, obtained with the 3 inhibitors (abemaciclib, palbociclib, ribociclib), differ in a statistically significant way (p<0.05) from each other (see Tables 1 and 2). To avoid repetition of data and to avoid overloading Tables 1 and 2, the authors chose not to include this information in the Tables again.

  1. A multivariate analysis should be performed.

Ans: This is a quite pertinent suggestion, indeed. The authors are grateful for this comment and have taken the opportunity to perform a multivariate logistic regression analysis to determine independent risk factors for seriousness to leukopenia and thrombocytopenia.

Accordingly, the authors created Table 3 which was added to the "Results", together with the following paragraph:

The seriousness of adverse drug reactions (leucopenia and thrombocytopenia) associated with CDK4/6 inhibitors (whether as monotherapy or in combination with concomitant drug therapy) was explored through a multivariate logistic regression (Table 3). Serious cases of leukopenia and thrombocytopenia were significantly more likely to occur for palbociclib and ribociclib when compared to abemaciclib (leukopenia - palbociclib: OR=146.220, 95%CI: 58.38-360.08; ribociclib: OR=50.568, 95%CI: 16.85-151.81; thrombocytopenia - palbociclib: OR=14.690, 95%CI: 9.13-23.63; ribociclib: OR=30.220, 95%CI: 8.99-101.57). When a CDK4/6 inhibitor was used in combination with another drug therapy (fulvestrant, letrozole, anastrozole, exemestane) the odds of serious cases of leukopenia decreased (OR=0.294, 95%CI: 0.14-0.62).

The statistical analysis carried out also led to the addition of point 3.3., entitled "Statistical Analysis", to the Materials and Methods:

Categorical variables were presented with their relative and absolute values. For statistical analysis Qui-square test was applied using the statistical software package SPSS 28.0 for Windows (SPSS. Chicago. IL. USA). A multivariate logistic regression analysis was performed to determine independent risk factors for seriousness to leukopenia and thrombocytopenia. P<0.05 indicates a statistically significant test result.

This statistical analysis and consequent interpretation and discussion of the results was carried out by a professor from the Department of Mathematics at the University of Beira Interior, who was also added to the authors. It should be noted that this mathematician, specialist in statistics, collaborated in the methodology, statistical validation and formal analysis as well as in the writing/review and editing of the revised manuscript.

Reviewer 3 Report

Dear Authors,

·        Review describes well about “CDK4/6 inhibitors induce their anti-cancer ac­tivity through the CD4/6-Rb axis, which is often disrupted in the majority of cancers and is at the basis of abnormal cell proliferation. When DNA synthesis occurs, CDK4/6 ki-nases, in the presence of mitogenic signals, bind to the D-type cyclins (cyclin D1, cyclin D2 and cyclin D3) and form catalytically active complexes.

·        One key function of these com­plexes is associated to the phosphorylation of the retinoblastoma tumor suppressor pro­tein (RB).

·        This protein is responsible for cell cycle transition from G1 to S phase. By pre­venting RB phosphorylation, CDK4/6 inhibitors block cell cycle progression”. Limitations of the this study -  data does not provide evidence on the causality relationship between the analyzed ADRs and the suspected drugs. 

·        Overall, Review’s structural information’s are good.                        

Moderate editing of english language required.

Author Response

Responses to Reviewers

The authors are grateful for comments and suggestions from all reviewers. All comments and suggestions have been taken into account, which has helped to improve the quality of the article. All changes to the article are marked in yellow.

The English has also been revised and improved. All these English corrections have not been reported so as not to unnecessarily burden the revised article with corrections.

Point-to-point responses to Reviewers

Reviewer 3

Review describes well about “CDK4/6 inhibitors induce their anti-cancer activity through the CD4/6-Rb axis, which is often disrupted in the majority of cancers and is at the basis of abnormal cell proliferation. When DNA synthesis occurs, CDK4/6 kinases, in the presence of mitogenic signals, bind to the D-type cyclins (cyclin D1, cyclin D2 and cyclin D3) and form catalytically active complexes.

Overall, Review’s structural information’s are good.

Ans: The authors are pleased with the interest the study has generated and are grateful for the summary of the main aspects of the article and the comments provided.

It should also be noted that the Methods, Results and Discussion were improved by carrying out a statistical analysis, using logistic regression, to determine independent risk factors for seriousness to leukopenia and thrombocytopenia.

Round 2

Reviewer 1 Report

The changes that have been made according to the reviewers comments make the mausctript acceptable for publication 

minor English editing